# Kinesin-8B controls basal body function and flagellum formation and is key to malaria transmission

Mohammad Zeeshan[1], David JP Ferguson[2], Steven Abel[3], Alana Burrrell[2], Edward Rea[1], Declan Brady[1], Emilie Daniel[1], Michael Delves[4], Sue Vaughan[2], Anthony A Holder[5], Karine G Le Roch[3], Carolyn A Moores[6], Rita Tewari[1]

**Eukaryotic flagella are conserved microtubule-based organelles that drive cell motility. *Plasmodium*, the causative agent of malaria, has a single flagellate stage: the male gamete in the mosquito. Three rounds of endomitotic division in male gametocyte together with an unusual mode of flagellum assembly rapidly produce eight motile gametes. These processes are tightly coordinated, but their regulation is poorly understood. To understand this important developmental stage, we studied the function and location of the microtubule-based motor kinesin-8B, using gene-targeting, electron microscopy, and live cell imaging. Deletion of the *kinesin-8B* gene showed no effect on mitosis but disrupted 9+2 axoneme assembly and flagellum formation during male gamete development and also completely ablated parasite transmission. Live cell imaging showed that kinesin-8B–GFP did not co-localise with kinetochores in the nucleus but instead revealed a dynamic, cytoplasmic localisation with the basal bodies and the assembling axoneme during flagellum formation. We, thus, uncovered an unexpected role for kinesin-8B in parasite flagellum formation that is vital for the parasite life cycle.**

## Introduction

Eukaryotic flagella, also known as motile cilia, are conserved microtubule (MT)-based organelles that protrude from cells and drive motility of single cells, or the movement of fluid across ciliated tissue (Mirvis et al, 2018). In a number of organisms, the mechanisms by which flagella are built and maintained are gradually being revealed, and defects in flagellum assembly and function are associated with numerous human diseases (Baker & Beales, 2009; Croft et al, 2018).

In general, flagella grow from mother centrioles—also known as basal bodies—which are recruited from the centrosome to act as templates for the characteristic ninefold symmetry of the MT array known as the flagellum axoneme (Mirvis et al, 2018). Specialised axonemal dynein motors are arranged according to the underlying axonemal MT organisation and these motors power flagella beating (Lin & Nicastro, 2018). Thus, flagella-driven motility depends on axonemal organisation defined by basal body function. However, the mechanisms that coordinate all these facets of flagella action are complex and less well understood.

Malaria is a disease caused by the unicellular parasite *Plasmodium* spp., which infects many vertebrates and is transmitted by female *Anopheles* mosquitoes (WHO, 2018). The parasite has a complex life cycle, with distinct asexual and sexual developmental stages. Extracellular, motile and invasive stages move by gliding motility powered by an actomyosin motor, except for male gametes, which use flagellar movement (Sinden et al, 1976, 2010; Boucher & Bosch, 2015; Frenal et al, 2017). Male gamete development, from the haploid gametocyte, is a very rapid process with endomitotic division and flagellum formation completed in 10–15 min (Sinden et al, 2010). Specifically, three successive rounds of DNA replication produce an 8N nucleus and are accompanied by synthesis and assembly of basal bodies and axonemes; then karyokinesis and exflagellation release eight haploid flagellated male gametes (Sinden et al, 1976).

The *Plasmodium* male gamete has a very simple structure with no subcellular organelles except an axoneme with associated dyneins, an elongated nucleus and a surrounding flagellar membrane (Sinden et al, 1976; Creasey et al, 1994; Okamoto et al, 2009). The axoneme—essential for flagellar motility—consists of a pair of central MTs (C1 and C2) encircled by nine doublet MTs—the so-called 9+2 organisation—as in many other organisms (Mitchell, 2004). However, the *Plasmodium* flagellum differs in several key respects in the mechanism of axoneme formation from that of other organisms including trypanosomes, *Chlamydomonas* and humans (Briggs et al, 2004; Bastin, 2010; Soares et al, 2019), as well as

[1]School of Life Sciences, Queens Medical Centre, University of Nottingham, Nottingham, UK  [2]Department of Biological and Medical Sciences, Faculty of Health and Life Science, Oxford Brookes University, Oxford, UK  [3]Department of Molecular, Cell and Systems Biology, University of California Riverside, Riverside, CA, USA  [4]London School of Hygiene and Tropical Medicine, Keppel, London, UK  [5]Malaria Parasitology Laboratory, Francis Crick Institute, London, UK  [6]Institute of Structural and Molecular Biology, Department of Biological Sciences, Birkbeck College, London, UK

Correspondence: rita.tewari@nottingham.ac.uk

other Apicomplexa with flagellated gametes such as *Toxoplasma* and *Eimeria* (Ferguson et al, 1974, 1977; Ferguson, 2009; Wilson et al, 2013). The differences in the mechanism of axoneme formation are (1) the *Plasmodium* basal body, from which the flagellum is assembled, does not exhibit the classical nine triplet MT (9+0) (Mirvis et al, 2018), but instead consists of an electron dense amorphous structure within which nine peripherally arranged MTs can sometimes be resolved (Forancia et al, 2015); (2) basal bodies/ centrioles are not present during asexual stages of the *Plasmodium* life cycle, unlike *Toxoplasma* and *Eimeria* where they play a role during asexual division (Ferguson & Dubremetz, 2014) but assemble de novo in male gametocytes (Sinden et al, 1976); (3) the axoneme itself is assembled from the basal bodies within the male gametocyte cytoplasm, and only after its assembly does the axoneme protrude from the membrane to form the flagellum itself; as a result and very unusually, axoneme construction occurs independently of intraflagellar transport (IFT) (Sinden et al, 1976; Briggs et al, 2004). Mechanisms that regulate *Plasmodium* male gamete biogenesis and control of axoneme length are largely unknown, although the involvement of some individual proteins has been described. For example, PF16 is a flagellar protein with an essential role in flagellar motility and the stability of the central MT pair in *Chlamydomonas* and *Plasmodium* (Smith & Lefebvre, 1996; Straschil et al, 2010), whereas SAS6, a well-conserved basal body protein, has a role in *Plasmodium* flagellum assembly (Nigg & Stearns, 2011; Marques et al, 2015). Proteomic analysis has revealed many potential regulators of axoneme assembly present in male gametes, including members of the kinesin superfamily (Talman et al, 2014). Kinesins are molecular motor proteins that have essential roles in intracellular transport, cell division, and motility (Wittmann et al, 2001; Verhey & Hammond, 2009; Cross & McAinsh, 2014). Some kinesin families are known to contribute to the ciliary structure and function, either by transporting ciliary components along axonemal MTs by IFT such as kinesin-2s (Scholey, 2008), or by destabilizing MTs, such as kinesin-9 and kinesin-13 (Blaineau et al, 2007; Dawson et al, 2007).

Kinesin-8s are conserved from protozoa to mammals, and can be subclassified as kinesin-8A, -8B and -8X (Wickstead et al, 2010; Vicente & Wordeman, 2015). In general, kinesin-8s are multitasking motors, able to move along MTs, cross-link and slide MTs, and influence MT dynamics at their ends (Mayr et al, 2007; Su et al, 2013). Kinesin-8As are best characterised as regulators of spindle length and chromosome positioning during metaphase (Straight et al, 1998; Savoian et al, 2004; Mary et al, 2015), whereas the mammalian kinesin-8B (Kif19) has a role in ciliary length control (Niwa et al, 2012).

The *Plasmodium* genome encodes two kinesin-8s (kinesin-8X and kinesin-8B), along with six or seven other kinesin proteins (Wickstead et al, 2010; Vicente & Wordeman, 2015; Zeeshan et al, 2019 *Preprint*). Kinesin-8X is conserved across the Apicomplexa, whereas kinesin-8B is restricted to certain genera such as *Toxoplasma*, *Eimeria*, and *Plasmodium* that have flagellated gametes (Zeeshan et al, 2019 *Preprint*). However, the role of kinesin-8s in regulation of *Plasmodium* flagella is unknown.

Here we have analysed the location and function of the kinesin-8B protein (PBANKA_0202700) present in male gametes in the rodent malaria parasite, *Plasmodium berghei*. Deletion of the gene

results in complete impairment of 9+2 axoneme formation because of misregulation of the basal body but has no effect on nuclear division. Its unprecedented involvement in basal body function prevents assembly of axonemes and the motile flagellum, and thereby blocks parasite transmission. Live cell imaging showed that kinesin-8B–GFP is expressed only during male gamete development and that, strikingly, it is not associated with the nuclear kinetochore but is present only in the cytoplasm. Here, we show that kinesin-8B–GFP is dynamically associated with the basal bodies and with axonemes during their formation and assembly, consistent with an unexpected role in flagellum genesis.

# Results

## Kinesin-8B is essential for male gamete formation and its deletion blocks parasite transmission

Based on its known expression in male gametes and a potential function during male gamete development, we first assessed the function of kinesin-8B throughout the *Plasmodium* life cycle by using a double crossover homologous recombination strategy to delete its gene in the context of a parasite line constitutively expressing GFP (Janse et al, 2006) (Fig S1A). This line expresses GFP constitutively in all stages of the parasite life cycle and is used routinely to facilitate phenotypic studies in both control and knockout lines. Diagnostic PCR confirmed successful integration of the targeting construct at the *kinesin-8B* locus (Fig S1B), and analysis by quantitative real time PCR (qRT-PCR) confirmed complete deletion of the *kinesin-8B* gene in this transgenic parasite (Fig S1C). Successful deletion of the *kinesin-8B* gene indicated that it is not essential for asexual blood stage development, consistent with the protein's expression and presence only during male gametogenesis, and with global functional studies (Bushell et al, 2017). Phenotypic analysis of the Δ*kinesin-8B* parasite was carried out at other developmental stages, comparing the wild-type (WT-GFP) parasite with two independent geneknockout parasite clones generated by two different transfections. Because Δ*kinesin-8B* parasites undergo asexual blood stage development, form gametocytes in mice, and exhibit no change in morphology or parasitemia, an essential role for kinesin-8B during these stages is unlikely. We then examined male and female gametocytes after activation in the exflagellation/ ookinete medium for the development and emergence of male and female gametes. This analysis revealed that there was no exflagellation in either of the Δ*kinesin-8B* parasite clones (clone 1 and clone 4), whereas exflagellation frequency for the WT-GFP parasites was normal (Fig 1A).

To investigate this defect further, we examined both zygote formation and ookinete development. No zygote (or ookinete) formation was detected in either clonal Δ*kinesin-8B* parasite line, whereas WT-GFP parasites showed normal zygote (ookinete) development (Fig 1B). To examine the effect of this defect on parasite transmission, *Anopheles stephensi* mosquitoes were fed on mice infected with Δ*kinesin-8B* parasites, and the number of oocsts on the mosquito gut was counted 14 d later. No oocysts were detected

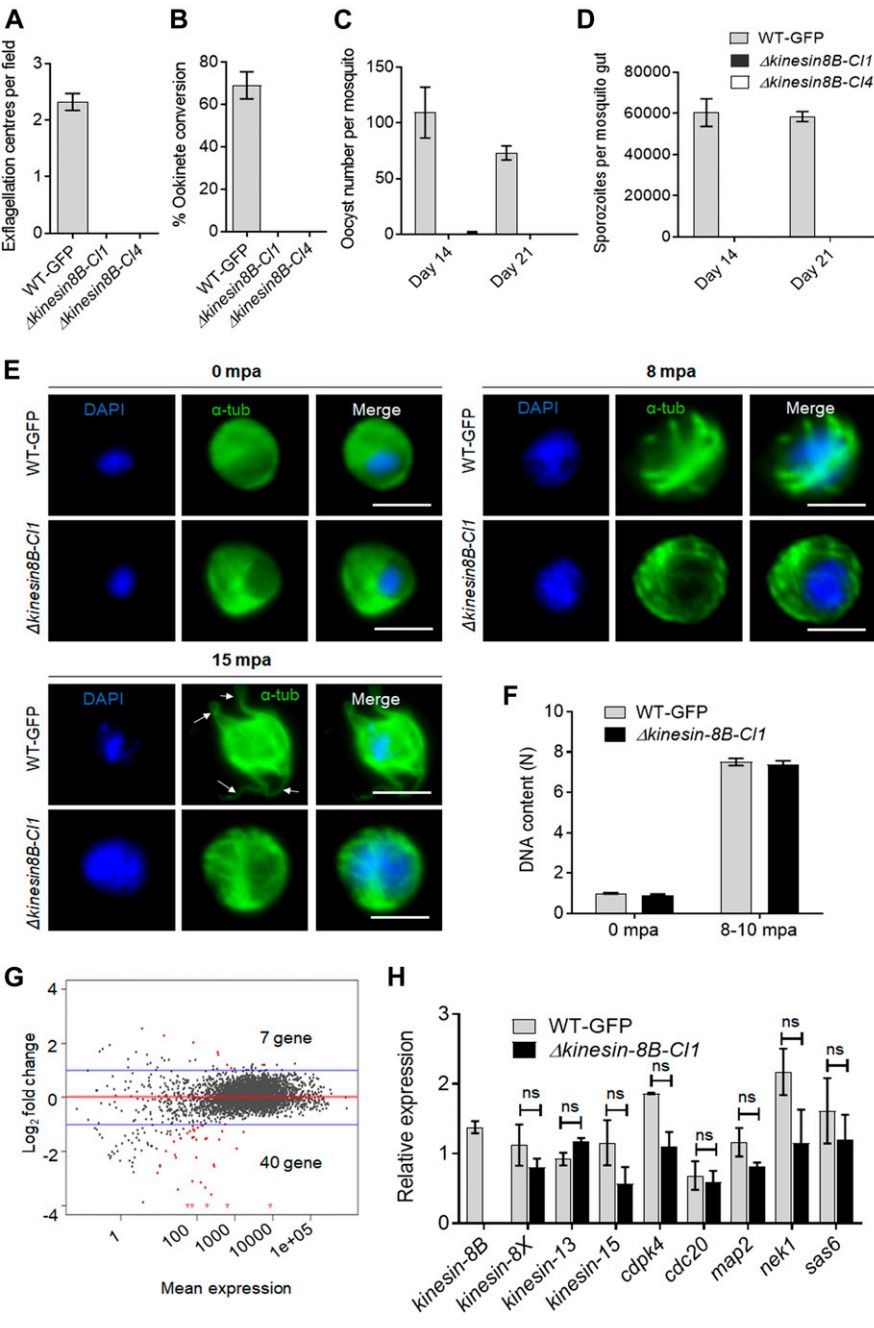

**Figure 1. Kinesin-8B is required for male gamete formation and parasite transmission.**
**(A)** Male gametes (determined by quantifying exflagellation centres) are absent in Δ*kinesin-8B* clones (Cl1 and Cl4) but not in wild-type parasites. Data are means from 15 random independent fields with a 40x objective and three independent replicates. Error bars: ± SEM. **(B)** Ookinete conversion from zygotes in wild-type and Δ*kinesin-8B* parasites (Cl1 and Cl4). n ≥ 3 independent experiments (>70 cells per experiment for wild type). Error bars: ± SEM. **(C)** Total number of oocysts of Δ*kinesin-8B* (Cl1 and Cl4) and WT-GFP parasites in mosquito midguts at 14 and 21 dpi. Data are shown as Mean ± SEM, n = 3 independent experiments. **(D)** Total number of sporozoites in oocysts of Δ*kinesin-8B* (Cl1 and Cl4) and WT-GFP parasites at 14 and 21 dpi. Mean ± SEM. n = 3 independent experiments. **(E)** In the absence of kinesin-8B, α-tubulin is not reorganised during gametogenesis. IFA images of α-tubulin in WT-GFP and Δ*kinesin-8B* male gametocytes either before activation (0 mpa) or 8 and 15 mpa. DAPI (DNA) is blue. White arrows indicate MT protrusions that occur as the flagellum develops in WT parasites, Scale bar = 5 μm. **(F)** Fluorometric analyses of DNA content (N) after DAPI nuclear staining. Male gametocytes were at 0 min (non-activated), or 8–10 mpa. The mean DNA content (and SEM) of 10 nuclei per sample are shown. Values are expressed relative to the average fluorescence intensity of 10 haploid ring-stage parasites from the same slide. **(G)** Differentially expressed genes in Δ*kinesin-8B* parasites compared with WT-GFP parasites. Samples were collected at 0 mpa. **(H)** qRT-PCR analysis of changes in transcription of selected genes affected in Δ*kinesin-8B* parasites. Error bars: ± SEM. The selected genes have an established or probable role in male gamete development. ns, non significant.

in mosquito guts infected with Δ*kinesin-8B* parasites, except on one occasion when we found four small oocysts in one mosquito after 14 d of infection with the clone 4 parasite line (Fig 1C). After 21 d of infection, no oocysts were observed for the Δ*kinesin-8B* lines, whereas the WT-GFP parasite line produced normal developing oocysts (Fig 1C). No sporozoites were found in the midguts of mosquitoes fed with Δ*kinesin-8B* parasites neither at 14 d post-infection (dpi) nor at 21 dpi (Fig 1D). To further confirm the lack of viable sporozoites, these mosquitoes were fed on naïve mice to assess parasite transmission. Mosquitoes were able to transmit the WT-GFP parasite and blood stage infection was observed in naïve

mice 4 d later. However, mosquitoes infected with Δ*kinesin-8B* parasites at the same time failed to transmit this parasite to susceptible mice (Fig S1D).

## Kinesin-8B is not required for DNA replication, but its deletion impairs male exflagellation

To understand the defect in more detail, we analysed the development of MT structures in Δ*kinesin-8B* and WT-GFP male gametocytes after activation in vitro by treatment with xanthurenic acid and decreased temperature at 0, 8, or 15 min postactivation (mpa).

We observed little evidence of early differences, but the MT distribution in the Δ*kinesin-8B* gametocytes gradually diverged from that of the WT-GFP parasites over time, and these mutants did not form flagella (as evidenced from tubulin labelling) at 15 mpa (arrow, Fig 1E). To assess the effect of the mutations on DNA replication during male gametogenesis, we analysed the DNA content (N) of Δ*kinesin-8B* and WT-GFP male gametocytes by fluorometric analyses after DAPI staining. We observed that Δ*kinesin-8B* male gametocytes had a haploid DNA content (1N) at 0 min (non-activated) and were octaploid (8N) 8–10 mpa, similar to WT-GFP parasites, indicating the absence of kinesin-8B had no effect on DNA replication (Fig 1F).

### Transcriptomic analysis identified no additional genes responsible for the Δ*kinesin-8B* phenotype

To investigate further the defect in male gamete development in Δ*kinesin-8B* parasites, we analysed the transcript profile of both non-activated and activated Δ*kinesin-8B* and control WT-GFP gametocytes. RNAseq analysis was performed on WT-GFP and Δ*kinesin-8B* gametocytes, just before activation (0 mpa) and after exflagellation (30 mpa) for four pairs of biological replicates (WT, 0 mpa; WT, 30 mpa; Δ*kinesin-8B*, 0 mpa; and Δ*kinesin-8B*, 30 mpa) and the read coverages exhibited Spearman correlation coefficients of 0.97, 0.98, 0.96, and 1.00, respectively, demonstrating the reproducibility of this experiment. As expected, no significant reads mapped to the *kinesin-8B* region in Δ*kinesin-8B* parasites (Fig S1E). Only 7 genes were up-regulated, and 40 genes were down-regulated compared with the WT-GFP control, but none was known to be important in gametogenesis (Fig 1G and Table S1). A qRT-PCR analysis of a specific set of genes coding for proteins identified either as molecular motors (kinesin family) or involved in male gamete development (Billker et al, 2004; Tewari et al, 2005) (Fig 1H) showed no significant change in transcript level. These data suggest that the observed phenotype resulted directly from the absence of kinesin-8B rather than through an indirect effect on another gene transcript.

### Activated Δ*kinesin-8B* gametocytes lack the 9+2 MT axoneme architecture

To examine ultrastructural differences between the Δ*kinesin-8B* and WT-GFP parasite lines during male gametogenesis, cells at 8, 15, and 30 mpa were examined by electron microscopy. The early developmental stages of both WT and mutant parasites contained a large central nucleus with diffuse chromatin (Fig 2Aa and b). As early as 8 mpa, elongated axonemes were identified running round the peripheral cytoplasm of the WT-GFP parasite (Fig2Aa and c). In cross section, the majority of axonemes (~60% based on examination of 36 random sections through microgametocytes) displayed nine peripheral doublet and two single central MTs; the classical 9+2 arrangement (Fig 2Ba and insert). However, even in the WT, a proportion of axonemes (~40%) show varying degrees of abnormality from loss of the central MTs to loss of a number of peripheral doublet MTs (Fig 2Ba, arrows). In contrast, an examination of 58 randomly sectioned Δ*kinesin-8B* microgametocytes failed to

identify a single 9+2 axoneme. The cytoplasm of the Δ*kinesin-8B* mutant contained numerous randomly oriented doublet and single MTs (Fig 2Ab and d). In cross section, these randomly distributed MTs showed little evidence of any organised interaction (Fig 2Bb and Inset). In longitudinally oriented sections, the MT growth appeared similar in both WT and mutant (Fig 2Ac and d) and the relative density of MTs also appeared similar (Fig 2Ba and b).

The nuclear appearance of mutant and WT-GFP parasites was similar in mid-stage microgametocyte development (Fig 2Ae and f); both had similar electron dense cone-shaped nuclear poles from which MTs radiated to form the nuclear spindle with attached kinetochores (Fig 2Ag and h). In WT parasites, an electron-dense basal body, which gives rise to the axoneme, could be identified adjacent to many nuclear poles. However, this relationship between nuclear pole and basal body was less obvious in the mutant. When quantified, it was observed that there were fewer basal bodies in the mutant (0.4/WT-GFP cell section compared with 0.3/mutant cell section based on examination of 25 and 58 cells, respectively). In addition, the association between the basal bodies and nuclear poles was dramatically reduced in Δ*kinesin-8B* mutant from 80% in WT compared with 21% in the mutant (cf. Fig 2Ag and h).

In the late stages of microgamete development (15 and 30 mpa), it was possible to observe chromatin condensation in the nuclei of WT-GFP and Δ*kinesin-8B* parasites, consistent with successful endomitosis in both (Fig 2Bc and d). In the WT-GFP parasite at 15 and 30 mpa, male gamete formation and exflagellation was observed, with the protrusion from the surface of the male gametocyte of a flagellum together with its attached nucleus leaving a residual body of cytoplasm (Fig 2Bc and e). Free male gametes consisting of an electron-dense nucleus and associated 9+2 flagellum were also observed (Fig 2Be and inset). In contrast, in the Δ*kinesin-8B* mutant at 15 and 30 mpa, there was little change in the cytoplasm, which still lacked any axonemal organisation but contained large numbers of randomly distributed MTs (Fig 2Bf and inset). No cytoplasmic processes or evidence of male gamete formation was observed.

In summary, deletion of the *kinesin-8B* gene did not appear to affect the initiation, growth, or density of single and doublet MTs but completely prevented their organisation into 9+2 axonemes. In addition, there was a reduction in the number of basal bodies and, more markedly, in their association with the nuclear poles. This defect ultimately prevented exflagellation and the formation of viable male gametes.

### Spatiotemporal profiles of kinesin-8B and the kinetochore protein Ndc80 show that kinesin-8B is absent from the mitotic spindle

To understand more precisely the role of kinesin-8B in building the male gametocyte flagellum, we investigated the subcellular location of kinesin-8B by live cell imaging in *P. berghei*. We generated a transgenic parasite line by single crossover recombination at the 3′ end of the endogenous *kinesin-8B* locus to express a C-terminal GFP-tagged fusion protein (Fig S2A). PCR analysis of genomic DNA using locus-specific diagnostic primers indicated correct integration of the GFP tagging construct (Fig S2B), and the presence of a protein of the expected size (~198 kD) in a gametocyte lysate was

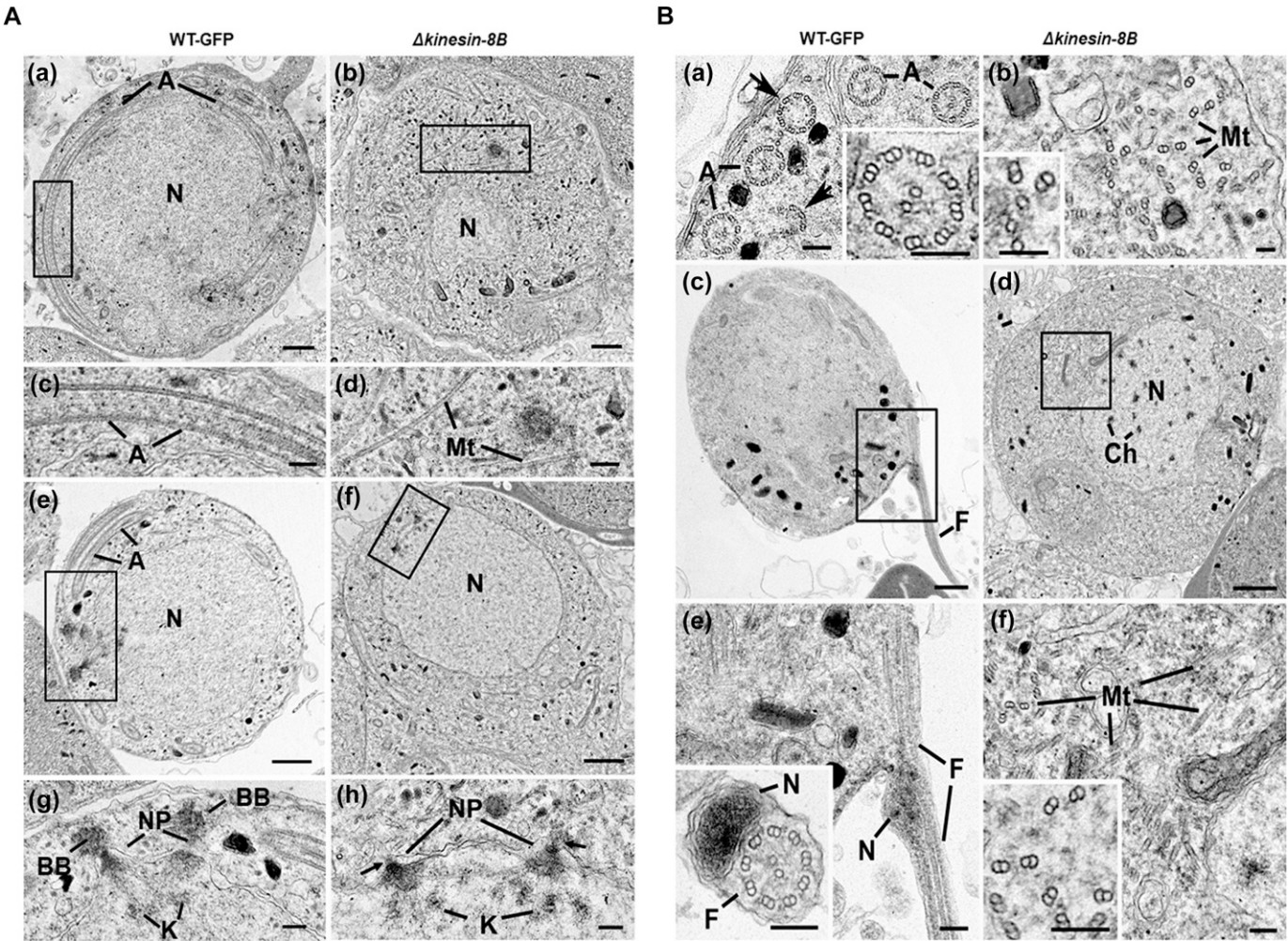

**Figure 2. Ultrastructure analysis reveals defects in basal body and axoneme formation.**

**(A)** Electron micrographs of mid-stage (8 mpa) male gametocytes of WT-GFP (a, c, e, and g) and Δ*kinesin-8B* (b, d, f, and h) parasites. Bars represent 1 µm in (a, b, e, and f) and 100 nm in (c, d, g, and h). (a) Low power image of a WT-GFP male gametocyte showing the large nucleus (N) with cytoplasm containing long axonemes (A) running around the periphery. (b) Low power image of a Δ*kinesin-8B* male gametocyte showing the nucleus (N) and the cytoplasm containing randomly orientated microtubules (Mt). (c) Detail of the enclosed area in panel (a) showing the parallel organization of the microtubules forming an axoneme (A). (d) Detail of the enclosed area in panel B showing randomly orientated microtubules (Mt). (e) WT-GFP male gametocyte showing the large nucleus (N) with multiple nuclear poles, spindle and basal bodies with cytoplasm containing a number of axonemes (A). (f) Δ*kinesin-8B* male gametocyte showing the nucleus with nuclear poles and spindle but no basal bodies. The cytoplasm lacks axonemes but has numerous microtubules (Mt). (g) Detail from the enclosed area in (e) showing basal bodies (BB), the nuclear pole (NP) connected by spindle microtubules with attached kinetochores (K). (h) Detail from the enclosed area in (f) showing the nuclear poles (NPs) connected by spindle microtubules with attached kinetochores (K). Note the absence of basal bodies associated with the nuclear poles (arrows). **(B)** Electron micrographs of mid (8 mpa) and late (15 mpa) male gametocytes of WT-GFP (a, c and e) and mutant (b, d and f). Bar represent 1 µm in (c and d) and 100 nm in all other micrographs. (a) Part of the peripheral cytoplasm of a mid-stage (8 mpa) WT-GFP gametocyte showing a cross section through a number 9+2 axonemes (A). Note the presence of incomplete axonemes (arrows). Inset. Detail of a cross sectioned axoneme showing the 9+2 organisation of the microtubules. (b) Part of the cytoplasm of a mid-stage mutant male gametocyte showing numerous randomly orientated microtubules (Mt). Inset. Detail showing the presence of both doublet and single microtubules. (c) Late stage (15 mpa) male gametocyte showing the partial formation of a male gamete by exflagellation. F, Flagellum. (d) Late mutant male gametocyte showing the early chromatin condensation and the absence of axonemes in the cytoplasm. (e) Detail of the enclose area in (c) showing the flagellum (F) and associated nucleus (N) protruding from the surface of the male gametocyte. Inset. Cross-section through a free male gamete showing the electron-dense nucleus (N) and the classical 9+2 flagellum. (f) Detail of the enclosed area in (d) showing the numerous randomly orientated microtubules (Mt). Inset. Detail showing the disorganisation of the doublet microtubules.

confirmed by Western blot analysis using GFP-specific antibody (Fig S2C). The expression and location of kinesin-8B was assessed by live cell imaging throughout the parasite life cycle; it was not observed in most stages except male gametocytes and gametes (Fig S3). This observation is consistent with the functional analysis indicating that kinesin-8B is essential for male gamete formation. The kinesin-8B–GFP parasites completed the full life cycle with no

detectable growth phenotypic defects resulting from the GFP tagging as our phenotypic analysis revealed throughout the parasite life cycle (Table S2).

To establish whether kinesin-8B is part of the mitotic assembly during the three rounds of chromosome replication in male gamete development, we examined its location together with that of the kinetochore protein Ndc80. Parasite lines expressing

kinesin-8B–GFP and Ndc80-cherry (manuscript in preparation) were crossed and used for live cell imaging of both markers to establish their spatiotemporal relationship. 1–2 min after gametocyte activation, kinesin-8B was observed close to the nucleus and adjacent to Ndc80, but not overlapping with it (Fig 3A). This pattern of adjacent location is also seen in later stages of development. This shows that both axoneme formation and chromosome division begin at a very early stage of gametogenesis and continue side by side (Fig 3A). Completion of both processes is coordinated in a spatiotemporal manner before the onset of exflagellation, but kinesin-8B's role is specifically related to axoneme formation in the cytoplasm.

## Kinesin-8B localises to cytoplasmic MTs and its movement is blocked by antimalarial compounds

To further explore the location of kinesin-8B, we investigated its co-localisation with MTs (using α-tubulin as a marker) by indirect immunofluorescence assay (IFA) using gametocytes fixed at different time points after activation. Kinesin-8B was localised on cytoplasmic MTs rather than on spindle MTs during male gamete formation. The pattern of both MTs and kinesin-8B was more distinct in later stages where their distribution adopted an axoneme-like pattern around the nucleus (Fig 3B). In addition, we tested two molecules with known antimalarial properties (TCMDC-123880 and

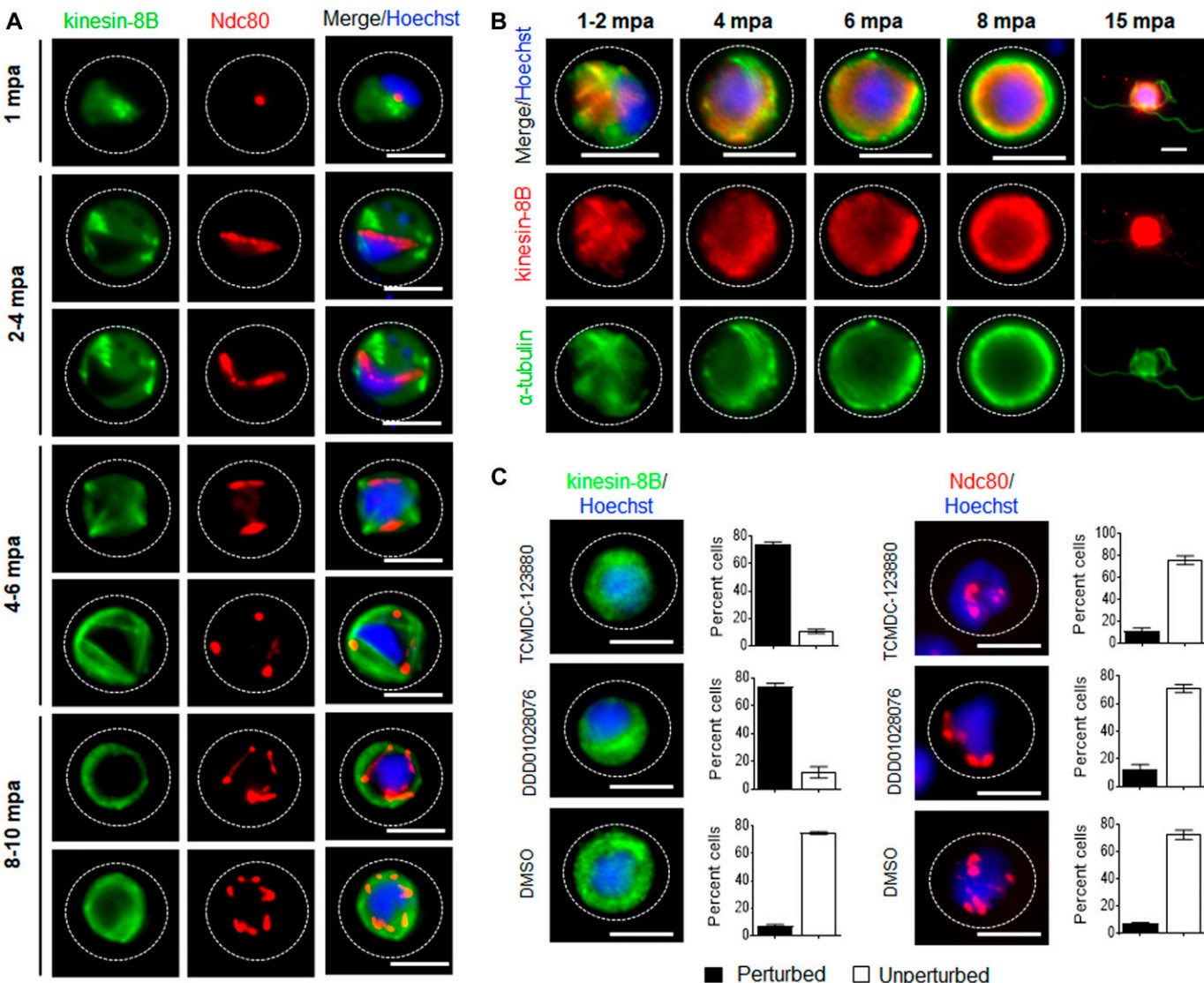

**Figure 3. The location of kinesin-8B in relation to that of the kinetochore (Ndc80) and microtubule (α-tubulin) markers.**
**(A)** The location of kinesin-8B–GFP (green) in relation to the kinetochore marker, Ndc80-mCherry (red) during male gamete formation. The cytoplasmic location of kinesin-8B contrasts with the nuclear location of Ndc80 during chromosome replication and segregation, indicating that kinesin-8B is not associated with the mitotic spindle. The non-clonal lines expressing kinesin-8B–GFP and Ndc80-mCherry were crossed and then individual parasites expressing both markers were analysed by live fluorescence microscopy. **(B)** Indirect IFA showing co-localisation of kinesin-8B (red) and α-tubulin (green) in male gametocytes 1 to 2, 4, 6, 8, and 15 mpa. **(C)** Antimalarial molecules block the dynamic distribution of kinesin-8B showing the resulting phenotype of compound addition at 4 mpa. In contrast to the effect on kinesin-8B–GFP distribution, no significant effect was seen on Ndc80–RFP. Inhibitors were added at 4 mpa and parasites were fixed at 8 mpa. Scale bar = 5 μm.

DDD01028076) with a putative role in targeting MT dynamics during male gamete development (Gamo et al, 2010; Delves et al, 2018). Addition of these molecules at 4 mpa blocked the dynamic distribution of kinesin-8B in more than 80% of male gametocytes but had no significant effect on Ndc80 dynamics (Fig 3C). This suggests that these compounds are most effective against cytoplasmic MT and serve to distinguish the cytoplasmic and nuclear MT-based processes.

### Kinesin-8B associates dynamically with basal bodies and growing axonemes

Because the Δ*kinesin-8B* parasite seemed to have defects in basal body interaction with nuclear pole and axoneme assembly, we analysed further the dynamic location of kinesin-8B by live imaging. We observed a diffuse cytoplasmic localisation in non-activated gametocytes and, after activation, kinesin-8B accumulated at one side of the nucleus, although retaining a cytoplasmic location. Within 1–2 min after activation, the distribution of kinesin-8B showed four clear foci close to the nucleus, suggesting its

association with the "tetrad of basal bodies," which is the template for axoneme polymerization (Fig 4A) (Sinden et al, 1976, 1978). These four foci duplicated further within 2–4 min to form another set of four kinesin-8B foci, which later moved apart from each other (Fig 4A). As gametogenesis proceeds, these kinesin-8B foci continued to move apart forming fibre-like structures around the nucleus by 4–6 mpa. The growth of these kinesin-8B–tagged fibres resembles the growing cytoplasmic axonemes (Marques et al, 2015; Straschil et al, 2010). The duplication of the four foci and emergence of these fibre-like structures occurred within 2–4 min (Fig 4B, Videos 1 and 2). By 6–8 mpa, these fibre-like structures decorated by kinesin-8B were arranged in a specific pattern with the basal bodies around the nucleus and forming a basket-like structure, similar to that described in earlier ultrastructural studies (Sinden et al, 1976, 1978). This structure may result from the completion of axoneme formation. Kinesin-8B was also observed with a continuous distribution along the length of the gametes as they emerged from the residual body of the gametocyte. Overall, the spatial and temporal profile of kinesin-8B revealed a localisation to basal bodies and axonemes throughout gametogenesis. Based on the ultrastructural

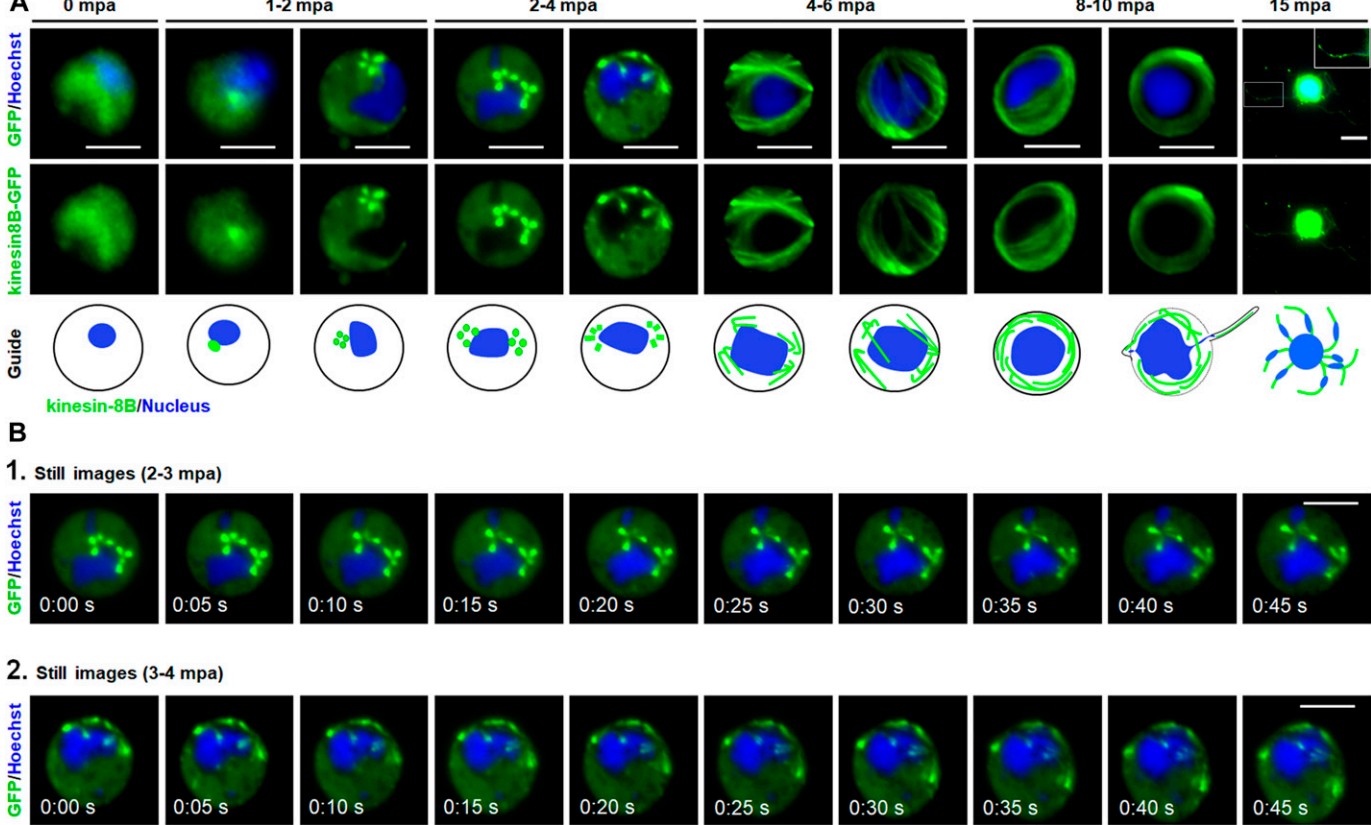

**Figure 4. The dynamic location of kinesin-8B in basal body and axoneme formation.**
**(A)** Cytoplasmic location of kinesin-8B–GFP (green) by live cell imaging in real time during male gamete development. DNA is stained with Hoechst 33,342 (blue). Before activation (0 min), kinesin-8B has a diffused location in male gametocyte cytoplasm. 1–2 mpa, it accumulates at one end of the nucleus and forms four foci reminiscent of the "tetrad of basal body" defined by electron microscopy (Sinden et al, 1976). These tetrad foci are duplicated within 2–4 mpa, and by 4–6 mpa fibre-like structures, representing axonemal growth decorated with kinesin-8B–GFP, extend from these tetrads to make a basket-like structure around the nucleus, which is completed by 8–10 mpa. After exflagellation, kinesin-8B–GFP is located along the length of the flagellum in the free male gamete (15 mpa, inset). For each time point, a cartoon guide is presented. **(B)** Still images (at every 5 s) of tetrad-foci duplication and the start of axonemal growth decorated with kinesin-8B–GFP at 2–3 mpa (Fig 4B1; Video 1) and 3–4 mpa (Fig 4B2; Video 2). Scale bar = 5 μm.

studies, live cell imaging, and immunofluorescence microscopy, we suggest a model for the dynamic location of kinesin-8B on basal bodies and axonemes during male gamete development, which is depicted in Fig 5.

## Discussion

Flagella and cilia, built around a central array of axonemal MT, are ancient organelles involved in motility and signalling (Carvalho-Santos et al, 2011; Soares et al, 2019). In some Apicomplexa parasites, including *Plasmodium,* only the male gamete is flagellated during the sexual stage of their complex life cycle, which determines transmission via the mosquito vector (Morrissette &

Sibley, 2002; Portman & Slapeta, 2014). *Plasmodium* male gametes are structurally very simple without any head or body attached to the flagellum. This highly distinctive flagella-driven movement (Wilson et al, 2013) presumably supports effective movement in the mosquito midgut. Flagella formation in *Plasmodium* spp. is unique compared with that of other eukaryotes, including the other closely related Apicomplexa, as they arise from de novo formed basal bodies and are directly assembled in the male gametocyte cytoplasm and, thus, do not require the building material to be transported by the IFT mechanisms of many other systems (Sinden et al, 1976, 2010; Briggs et al, 2004). This assembly mechanism is rapid and, again, is presumably an adaptation to facilitate efficient transition through this life cycle stage in the mosquito gut.

We show that *Plasmodium* kinesin-8B has a specific role in building the male gamete flagellum, and, in its absence, parasite transmission from insect vector to mammalian host is blocked. The severe defect in exflagellation exhibited by Δ*kinesin-8B* parasites is consistent with the location of the protein we describe and with its presence in the male proteome (Khan et al, 2005; Talman et al, 2014). Furthermore, a specific role for kinesin-8B at this life cycle stage is consistent with it being dispensable during blood stages (this study and Bushell et al (2017)).

Male gamete development in the malaria parasite involves three rounds of DNA replication and mitotic division followed by chromosome condensation and exflagellation, resulting in eight gametes (Sinden et al, 2010; Guttery et al, 2012). DNA content analysis of the Δ*kinesin-8B* mutant showed that the normal three rounds of replication occurred since octaploid nuclei (8N) were observed at 8–10 mpa. The distinct separation of kinesin-8B from the kinetochore marker Ndc80 and the normal nuclear ultrastructure further substantiates the idea that this kinesin is not part of the mitotic spindle within the nucleus. The experiments using *Plasmodium*-specific inhibitors (Gamo et al, 2010; Delves et al, 2018) provide further discrimination between MT-dependent events in the nucleus—where these compounds have no effect on Ndc80 localisation and, therefore, presumably no effect on mitotic spindle MTs—and the cytoplasm, where they have considerable effect on cytoplasmic axoneme assembly and kinesin-8B localisation. This functional discrimination is further supported by a recent study, which shows that deletion of $\alpha$-1 tubulin does not affect genome replication and nuclear division but results in oocyst in which no MTs are detected (Spreng et al, 2019). There is also evidence that although kinesin-8X of *P. berghei* is associated with nuclear spindle formation, its deletion does not prevent microgamete formation (Zeeshan et al, 2019 *Preprint*). These findings suggest a distinct molecular makeup and functionality of different MT arrays in *Plasmodium* parasites.

Global transcript analysis and qRT-PCR examination of genes coding for proteins shown to affect cytokinesis and chromosome condensation, such as CDPK4, CDC20, GEST, and MAP2 (Billker et al, 2004; Tewari et al, 2005; Talman et al, 2011; Guttery et al, 2012) revealed no differentially expressed genes in the defective Δ*kinesin-8B* parasite that were deemed important for male gametogenesis. This suggests that the observed phenotype is due to the absence of kinesin-8B alone.

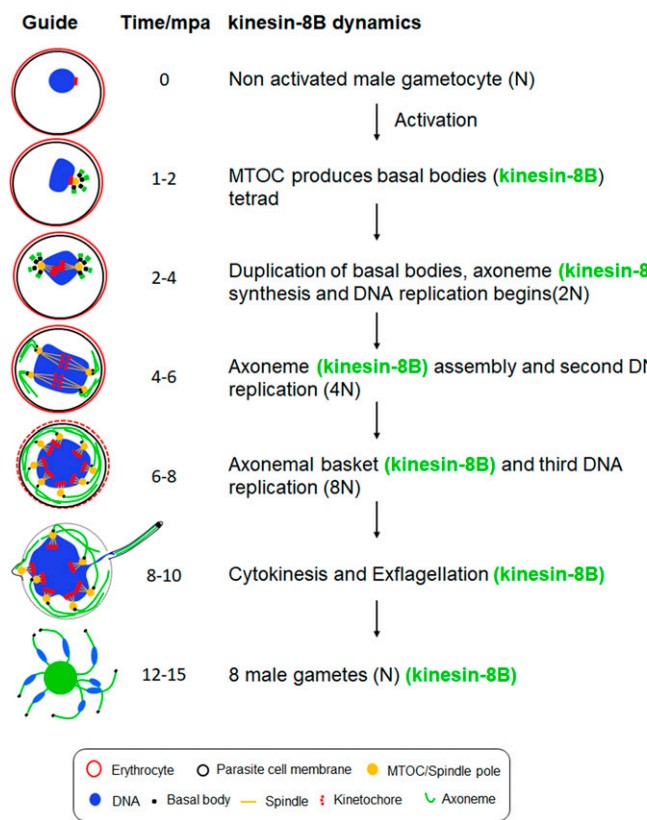

**Figure 5. Schematic model of kinesin-8B location on basal bodies and axonemes assembly during the male gamete development.**
Kinesin-8B accumulates at one end of the nucleus within 1 mpa of male gametocytes, probably near the MT organizing centre (MTOC) and develops into four foci corresponding to the tetrad of basal bodies (BBs, green). These tetrad foci of kinesin-8B duplicate and separate from each other within 2–4 mpa. Each of these foci is a basal body that serves as a platform for axonemal MT assembly. The fibre-like structure decorated with kinesin-8B (green) emerges from these foci, showing their association with axoneme assembly. Meanwhile, DNA replication accompanies basal body duplication, with associated mitotic spindles (yellow) and kinetochores (red) via the spindle pole (Sinden et al, 1976). The axonemal fibre-like structures develop further into a basket-like structure around the nucleus by 6–10 mpa. Axoneme formation and DNA replication are completed by this time (8–10 mpa) and a haploid genome connected with a basal body, is pulled into an emerging gamete.

Our ultrastructure studies provide a very striking comparison between the assembled axonemes of wild-type parasites and the lack of proper axoneme assembly in the absence of kinesin-8B. Examination of three different time points (8, 15, and 30 mpa) revealed that single and doublet MTs were formed and elongated in the mutant parasites, but they were not organised into the classical 9+2 symmetry. Furthermore, although the amorphous nature of the basal body precluded identification of structural abnormalities in the absence of kinesin-8B, there appeared to be at least partial loss of the tight association between the basal body and the nuclear pole. This phenotype is very different from that of many other mutants with exflagellation defects (Billker et al, 2004; Ferguson & Dubremetz, 2014; Wall et al, 2018), but there is some similarity to that of other mutants affecting the basal body (SAS6) or involved in assembly of the central pair of MTs (PF16) in *Plasmodium* axonemes (Straschil et al, 2010; Marques et al, 2015). These data suggest that kinesin-8B deletion causes a defect in both the basal body and axonemes.

Using live cell imaging, we show the dynamic localisation of kinesin-8B from the earliest stages of male gamete development (summarized in Fig 5). Kinesin-8B is a cytoplasmic protein exhibiting a basal body-like pattern in the early stages of gamete development—this highlights the fact that basal body replication occurs within 2 min of activation and that associated axoneme-like structures are assembled later. The association of kinesin-8B with axonemes is supported by co-localisation with cytoplasmic MT. The pattern of axoneme development revealed by tracking kinesin-8B is consistent with earlier ultrastructural studies (Sinden et al, 1976, 1978). The formation of basal bodies from a single MTOC-like structure was evident at a very early time in gamete development (Sinden et al, 1976, 1978). Here, we have displayed the real-time dynamics of their duplication and localisation as distinct structures. These features are consistent with the basal body defects resulting in the disorganised axoneme assembly that we observe with the deletion of *kinesin-8B*.

Kinesin motors participate in diverse functions in many processes, including cell division, intracellular transport, IFT, and controlling MT dynamics (Dawson et al, 2007; Scholey, 2008; Verhey & Hammond, 2009; Cross & McAinsh, 2014). The lack of association of *P. berghei* kinesin-8B with mitotic assembly within the nucleus—and with kinetochores in particular—is distinct from what is seen for kinesin-8As in most eukaryotes (Savoian et al, 2004; Savoian & Glover, 2010; Wang et al, 2016). The metazoan kinesin-8B, Kif19, is involved in ciliary length regulation, but the role of kinesin-8B in axonemal assembly seems to be more profound in *Plasmodium* than controlling length alone. Kinesin-8B is consistently present in those Apicomplexa with flagellated gametes such as *Plasmodium*, *Toxoplasma*, and *Eimeria* and absent in other genera such as *Theileria*, *Babesia*, and *Cryptosporidium*, which lack flagellated gametes (Zeeshan et al, 2019 Preprint). The properties of kinesin-8B contrast with those of *Plasmodium* kinesin-8X, which is located on the mitotic spindle (Zeeshan et al, 2019 Preprint).

In conclusion, this is the first study exploring the real-time dynamics and functional role of the kinesin-8B molecular motor in flagellum formation during *Plasmodium* male gamete development. It has a key role in establishing basal body formation and

axoneme structure and assembly, thereby regulating male gamete development, which is an essential stage in parasite transmission.

# Materials and Methods

### Ethics statement

The animal work performed in this study passed an ethical review process and was approved by the U.K. Home Office. Work was carried out under U.K. Home Office Project Licenses (40/3344 and 30/3248) in accordance with the U.K. "Animals (Scientific Procedures) Act 1986." 6–8-wk-old female Tuck ordinary (Harlan) outbred mice were used for all experiments in the United Kingdom.

### Generation of transgenic parasites

The gene-deletion targeting vector for kinesin-8B (PBANKA_020270) was constructed using the pBS-DHFR plasmid, which contains polylinker sites flanking a *Toxoplasma gondii* dhfr/ts expression cassette conferring resistance to pyrimethamine, as described previously (Saini et al, 2017). PCR primers N1261 and N1262 were used to generate a 1,000-bp fragment of kinesin-8B 5′ upstream sequence from genomic DNA, which was inserted into *Apa*I and *Hin*dIII restriction sites upstream of the dhfr/ts cassette of pBS-DHFR. A 1,240-bp fragment generated with primers N1263 and N1264 from the 3′ flanking region of *kinesin-8B* was then inserted downstream of the dhfr/ts cassette using *Eco*RI and *Xba*I restriction sites. The linear targeting sequence was released using *Apa*I/*Xba*I. WT-GFP (ANKA line WTGFPcon 507cl1) expressing constitutively GFP (Janse et al, 2006) was used for transfection. A schematic representation of the endogenous *Pbkinesin-8B* locus, the constructs, and the recombined *kinesin-8B* locus, can be found in Fig S1.

The C terminus of kinesin-8B was tagged with GFP by single crossover homologous recombination in the parasite. To generate the kinesin-8B–GFP line, a region of the *kinesin-8B* gene downstream of the ATG start codon was amplified using primers T1991 and T1992, ligated to p277 vector, and transfected as described previously (Guttery et al, 2012). A schematic representation of the endogenous *kinesin-8B* locus (PBANKA_020270), the constructs, and the recombined *kinesin-8B* locus can be found in Fig S2. The oligonucleotides used to generate the mutant parasite lines can be found in Table S3. *P. berghei* ANKA line 2.34 (that does not express any GFP) was transfected by electroporation to generate the kinesin-8B–GFP transgenic line (Janse et al, 2006).

### Parasite genotype analyses

For the gene knockout parasites, diagnostic PCR was used with primer 1 (IntN126) and primer 2 (ol248) to confirm integration of the targeting construct, and primer 3 (N102 KO1) and primer 4 (N102 KO2) were used to confirm deletion of the *kinesin-8B* gene (Fig S1). For the parasites expressing a C-terminal GFP-tagged kinesin-8B protein, diagnostic PCR was used with primer 1 (IntT199) and primer 2 (ol492) to confirm integration of the GFP targeting construct (Fig S2).

## Purification of gametocytes

The purification of gametocytes was achieved using a protocol described previously (Beetsma et al, 1998) with some modifications. Briefly, parasites were injected into phenylhydrazine-treated mice and enriched by sulfadiazine treatment after 2 d of infection. The blood was collected on day 4 after infection, and gametocyte-infected cells were purified on a 48% vol/vol NycoDenz (in PBS) gradient (NycoDenz stock solution: 27.6% wt/vol NycoDenz in 5 mM Tris–HCl, pH 7.20, 3 mM KCl, and 0.3 mM EDTA). The gametocytes were harvested from the interface and washed.

## Live cell and time-lapse imaging

Purified gametocytes were examined for GFP expression and localisation at different time points (0, 1–15 min) after activation in ookinete medium (Guttery et al, 2014). Images were captured using a 63× oil immersion objective on a Zeiss Axio Imager M2 microscope fitted with an AxioCam ICc1 digital camera (Carl Zeiss, Inc). Time-lapse videos (1 frame every 5 s for 15–20 cycles) were taken with a 63× objective lens on the same microscope and analysed with the AxioVision 4.8.2 software.

## Fixed IFA and measurements

The purified gametocytes from kinesin-8B–GFP, WT-GFP, and Δ*kinesin-8B* parasites were activated in the ookinete medium and then fixed at 0 min, 1–2 min, 6–8 min, and 15 min postactivation with 4% PFA (Sigma-Aldrich) diluted in microtubule stabilising buffer for 10–15 min and added to poly-L-lysine–coated slides. Immunocytochemistry was performed using primary GFP-specific rabbit monoclonal antibody (mAb) (A1122; Invitrogen; used at 1:250) and primary mouse anti-*α* tubulin mAb (T9026; Sigma; used at 1:1,000). Secondary antibodies were Alexa 488–conjugated antimouse IgG (A11004; Invitrogen) and Alexa 568–conjugated antirabbit IgG (A11034; Invitrogen) (used at 1 in 1,000). The slides were then mounted in Vectashield 19 with DAPI (Vector Labs) for fluorescence microscopy. Parasites were visualised on a Zeiss Axio Imager M2 microscope fitted with an AxioCam ICc1 digital camera (Carl Zeiss, Inc).

To measure nuclear DNA content of activated microgametocytes by direct immunofluorescence, images of parasites fixed (0 mpa and 8–10 mpa) and stained as above were analysed using the ImageJ software (version 1.44) (National Institute of Health) as previously described (Tewari et al, 2005).

## Generation of dual tagged parasite lines

The kinesin-8B–GFP parasites were mixed with Ndc80-cherry parasites in equal numbers and injected into a mouse. Mosquitoes were fed on this mouse 4–5 d after infection when gametocyte parasitemia was high. These mosquitoes were checked for oocyst development and sporozoite formation at day 14 and day 21 after feeding. Infected mosquitoes were then allowed to feed on naïve mice, and after 4–5 d, these mice were examined for blood stage parasitemia by microscopy with Giemsa-stained blood smears. In this way, some parasites expressed both kinesin-8B–GFP and Ndc80-cherry in the resultant gametocytes, and these were purified and fluorescence microscopy images were collected as described above.

## Inhibitor studies

Gametocytes were purified as above and treated with two anti-malarial molecules (TCMDC-123880 and DDD01028076) (Gamo et al, 2010; Delves et al, 2018) at 4 mpa and then fixed with 4% PFA at 8 min after activation. DMSO was used as a control treatment. These fixed gametocytes were then examined on a Zeiss Axio Imager M2 microscope fitted with an AxioCam ICc1 digital camera (Carl Zeiss, Inc).

## Parasite phenotype analyses

Blood containing approximately 50,000 parasites of the Δ*kinesin-8B* line was injected intraperitoneally (i.p.) into mice to initiate infections. Asexual stages and gametocyte production were monitored by microscopy on Giemsa-stained thin smears. 4–5 d postinfection, exflagellation and ookinete conversion were examined as described previously (Guttery et al, 2012) with a Zeiss Axio Imager M2 microscope (Carl Zeiss, Inc) fitted with an AxioCam ICc1 digital camera. To analyse mosquito transmission, 30–50 *Anopheles stephensi* SD 500 mosquitoes were allowed to feed for 20 min on anaesthetized, infected mice with an asexual parasitemia of 15% and a comparable number of gametocytes as determined on Giemsa-stained blood films. To assess midgut infection, ~15 guts were dissected from mosquitoes on day 14 postfeeding, and oocysts were counted on an AxioCam ICc1 digital camera fitted to a Zeiss Axio Imager M2 microscope using a 63× oil immersion objective. On day 21 postfeeding, another 20 mosquitoes were dissected, and their guts crushed in a loosely fitting homogenizer to release sporozoites, which were then quantified using a haemocytometer or used for imaging. Mosquito bite back experiments were performed 21 d postfeeding using naive mice, and blood smears were examined after 3–4 d.

## Electron microscopy

Gametocytes activated for 8, 15, and 30 min were fixed in 4% glutaraldehyde in 0.1 M phosphate buffer and processed for electron microscopy as previously described (Ferguson et al, 2005). Briefly, the samples were postfixed in osmium tetroxide, treated *en bloc* with uranyl acetate, dehydrated, and embedded in Spurr epoxy resin. Thin sections were stained with uranyl acetate and lead citrate before examination in a Hitachi H-7560 electron microscope (Hitachi UK Ltd).

## qRT-PCR analyses

RNA was isolated from gametocytes using an RNA purification kit (Stratagene). cDNA was synthesised using an RNA-to-cDNA kit (Applied Biosystems). Gene expression was quantified from 80 ng of total RNA using an SYBR green fast master mix kit (Applied Biosystems). All the primers were designed using the Primer3 software (Primer-blast; NCBI). Analysis was conducted using an Applied Biosystems 7500 fast machine with the following cycling conditions:

95°C for 20 s followed by 40 cycles of 95°C for 3 s; 60°C for 30 s. Three technical replicates and three biological replicates were performed for each assayed gene. The *hsp70* (PBANKA_081890) and *arginyl-t RNA synthetase* (PBANKA_143420) genes were used as endogenous control reference genes. The primers used for qPCR can be found in Table S3.

### RNAseq analysis

Libraries were prepared from lyophilized total RNA using the KAPA Library Preparation Kit (KAPA Biosystems) after amplification for a total of 12 PCR cycles (12 cycles [15 s at 98°C, 30 s at 55°C, and 30 s at 62°C]) using the KAPA HiFi HotStart Ready Mix (KAPA Biosystems). Libraries were sequenced using a NextSeq500 DNA sequencer (Illumina), producing paired-end 75-bp reads.

FastQC (https://www.bioinformatics.babraham.ac.uk/projects/fastqc/), was used to analyse raw read quality, and based on this information, the first 10 bp of each read and any adapter sequences were removed using Trimmomatic (http://www.usadellab.org/cms/?page=trimmomatic). The resulting reads were mapped against the *P. berghei* ANKA genome (v36) using HISAT2 (version 2-2.1.0), using default parameters. Uniquely mapped, properly paired reads were retained using SAMtools (http://samtools.sourceforge.net/). Genome browser tracks were generated and viewed using the Integrative Genomic Viewer (Broad Institute).

Raw read counts were determined for each gene in the *P. berghei* genome using BedTools (https://bedtools.readthedocs.io/en/latest/#) to intersect the aligned reads with the genome annotation. Differential expression analysis was performed using DESeq2 to calculate $\log_2$-fold expression changes between the knockout and wild-type conditions for every gene. Gene ontology enrichment was perfumed either using R package TopGO, or on PlasmoDB (http://plasmodb.org/plasmo/) with repetitive terms removed by REVIGO (http://revigo.irb.hr/).

### Statistical analysis

All statistical analyses were performed using GraphPad Prism 7 (GraphPad Software). For qRT-PCR, an unpaired *t* test was used to examine significant differences between wild-type and mutant strains.

## Data Availability

Sequence reads have been deposited in the NCBI Sequence Read Archive with accession number: PRJNA549466.

## Supplementary Information

## Acknowledgements

We thank Julie Rodgers for helping in maintaining the insectary and other technical works. We also thank Dr Antonio Mendes and other staff of the insectary at Lisbon for helping us with our mosquito colony. This work was supported by Medical Research Council UK (G0900278, MR/K011782/1, and MR/N023048/1) and Biotechnology and Biological Sciences Research Council (BB/N017609/1) to R Tewari and M Zeeshan; the BBSRC (BB/N018176/1) to CA Moores; the Francis Crick Institute (FC001097), the Cancer Research UK (FC001097), the UK Medical Research Council (FC001097), and the Wellcome Trust (FC001097) to AA Holder; and the National Institute of Health/National Institute of Allergy and Infectious Disease (R01 AiI136511) and the University of California, Riverside (NIFA-Hatch-225935) to KG Le Roch.

### Author Contributions

M Zeeshan: formal analysis, validation, investigation, visualization, methodology, and writing—original draft, review, and editing.
DJP Ferguson: conceptualization, data curation, software, formal analysis, supervision, validation, investigation, visualization, methodology, and writing—original draft, review, and editing.
S Abel: formal analysis and methodology.
A Burrrell: formal analysis and methodology.
E Rea: formal analysis and methodology.
D Brady: formal analysis, validation, and methodology.
E Daniel: formal analysis and methodology.
M Delves: formal analysis and methodology.
S Vaughan: resources and methodology.
AA Holder: conceptualization, supervision, funding acquisition, visualization, and writing—review and editing.
KG Le Roch: formal analysis, funding acquisition, investigation, methodology, and writing—review and editing.
CA Moores: conceptualization, data curation, formal analysis, supervision, funding acquisition, and writing—review and editing.
R Tewari: conceptualization, resources, data curation, formal analysis, supervision, funding acquisition, validation, investigation, visualization, methodology, and writing—original draft, review, and editing.

### Conflict of Interest Statement

The authors declare that they have no conflict of interest.

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
