## [Reviewer comments · Life Science Alliance]

Life Science Alliance

Kinesin-8B controls basal body function and flagellum formation and is key to malaria transmission

Rita Tewari, Mohammad Zeeshan, Stevel Abel, Alana Burrell, Edward Rea, Declan Brady, Emilie Daniel, Michael Delves, Tony Holder, Karine Le Roch, Carolyn Moores, Sue Vaughan, and David Ferguson

DOI: <https://doi.org/10.26508/lsa.201900488>

Corresponding author(s): Rita Tewari, University of Nottingham

Review Timeline:

Submission Date:	2019-07-17
Editorial Decision:	2019-07-23
Revision Received:	2019-08-01
Editorial Decision:	2019-08-02
Revision Received:	2019-08-05
Accepted:	2019-08-06

Scientific Editor: Andrea Leibfried

Transaction Report:

July 23, 2019

Re: Life Science Alliance manuscript #LSA-2019-00488-T

Dr. Rita Tewari
University of Nottingham
School of Life Sciences
Queens Medical Centre
Nottingham NG7 2UH
United Kingdom

Dear Dr. Tewari,

Thank you for submitting your manuscript entitled "Kinesin-8B controls basal body function and flagellum formation and is key to malaria parasite transmission" to Life Science Alliance. The manuscript was assessed by expert reviewers, whose comments are appended to this letter.

As you will see, the reviewers appreciate your findings and provide constructive input on how to further strengthen your work. We would thus like to invite you to submit a revised version of your manuscript, addressing the points raised by the reviewers. The revision requests seem minor and straightforward to address, but please do get in touch in case you would like to discuss an individual point further.

Thank you for this interesting contribution to Life Science Alliance. We are looking forward to receiving your revised manuscript.

Sincerely,

B. MANUSCRIPT ORGANIZATION AND FORMATTING:

Reviewer #1 (Comments to the Authors (Required)):

The is a very interesting study in which the authors have undertaken a very thorough analysis of the organisation and dynamics of nuclear and cytoplasmic microtubule dynamics during male gametogenesis and the potential role of microtubule-based motor, kinesin-8B.

The authors examined the effect of disruption of the gene encoding the kinesin-8B, in *P. berghei*. They report that disruption of kinesin-8B has no effects in asexual stages, as anticipated. Interestingly, it caused disruption of 9+2 axoneme assembly and flagellum formation during male gamete development.

The authors show that the defect appears to arise due to a disrupted interaction between the basal bodies and nuclear poles, suggesting kinesin-8B play a role in connecting the basal body to the MTOC.

The authors used transfectants in which the C-terminus of kinesin-8B was tagged with GFP. They were able to follow the dynamics of kinesin-8B revealing that it is associated with the basal bodies and the growing axonemes, rather than with nuclear microtubules.

This is an important study with surprising findings and the results are described very clearly.

The only potential issue is that the authors used the transfectants in which the C-terminus of kinesin-8B was tagged with GFP as their control line (WT-GFP) for comparison with the kinesin-8B-disrupted line. There is no discussion of whether the GFP tagging might affect the normal function of kinesin-8B. This is important given that some abnormal and incomplete axonemes were observed (Figure 2.2a) in the WT-GFP line. It would be useful to include a comparison with a line in which the kinesin-8B is not tagged.

Reviewer #2 (Comments to the Authors (Required)):

The authors describe the functional involvement of kinesin 8 B in formation of axonemes in *Plasmodium* by gene deletion and gene tagging.

As such this is a straightforward and well written and nicely illustrated paper that could in principle be published as is. However, some small additions would really improve the paper. I would have indicated some minor spellings as well if there were line numbers.

Is the kinesin-GFP only visible in sexual stages or also in other stages along the life cycle? If yes, please show, if just in supplement.

If the authors have electron microscopy data from later time points (e.g. 30 minutes) on the mutants I would encourage them to show this. Maybe flagella do assemble to some degree and the RNAseq analysis was performed at this time.

Can the authors quantify the amount of microtubules in the KO vs the WT? It appears that the microtubules are just as long in the KO as in the WT and that there are just as many microtubules present, is that impression correct?

What is the localization of Ndc80 in other parasites stages (this might be part of a separate paper and could be stated as such, alternatively a figure with different life cycle stages would be informative).

It should be clearly stated in the figure legend if these parasites were not clonal lines.

Some references are missing page or e or doi numbers

The discussion is not very eloquent and could be improved by subheadings and including a small discussion on two different aspects:

1. Motility of the gametes: Plasmodium gametes move in a very distinct way from other flagella and only one paper has really examined this in detail (Wilson et al., PNAS 2010). This would deserve mention and some discussion as I think no paper on gametes can be complete without this landmark study (could also be mentioned in the introduction)

2. Dissociation of nuclear division with gamete formation is reminiscent of a similar phenotype observed after microtubule depletion in oocysts, where nuclear division still occurs (albeit in the absence of microtubules) but no sporozoites are formed (Spreng et al., EMBO J 2019). This, together with the authors observation on kinesin 8B and the two drugs suggests a distinct molecular makeup and functionality of the different microtubule arrays in the parasite.

Reviewers' comments with responses (In blue)

We thank both reviewers for their positive and constructive comments and their suggestions to improve the manuscript, which we have carefully addressed, including the insertion of additional references. We hope that these revisions satisfy all the reviewers' queries and concerns. We provide below our responses (highlighted in blue) to specific points made by the reviewers:

Reviewer #1 (Comments to the Authors (Required)):

The is a very interesting study in which the authors have undertaken a very thorough analysis of the organisation and dynamics of nuclear and cytoplasmic microtubule dynamics during male gametogenesis and the potential role of microtubule-based motor, kinesin-8B.

The authors examined the effect of disruption of the gene encoding the kinesin-8B, in *P. berghei*. They report that disruption of kinesin-8B has no effects in asexual stages, as anticipated. Interestingly, it caused disruption of 9+2 axoneme assembly and flagellum formation during male gamete development.

The authors show that the defect appears to arise due to a disrupted interaction between the basal bodies and nuclear poles, suggesting kinesin-8B play a role in connecting the basal body to the MTOC.

The authors used transfectants in which the C-terminus of kinesin-8B was tagged with GFP. They were able to follow the dynamics of kinesin-8B revealing that it is associated with the basal bodies and the growing axonemes, rather than with nuclear microtubules.

This is an important study with surprising findings and the results are described very clearly.

We are very pleased that the reviewer considers this an interesting, thorough and important study, and we thank them for their valuable and constructive comments.

The only potential issue is that the authors used the transfectants in which the C-terminus of kinesin-8B was tagged with GFP as their control line (WT-GFP) for comparison with the kinesin-8B-disrupted line. There is no discussion of whether the GFP tagging might affect the normal function of kinesin-8B. This is important given that some abnormal and incomplete axonemes were observed (Figure 2.2a) in the WT-GFP line. It would be useful to include a comparison with a line in which the kinesin-8B is not tagged.

To clarify, we used two control parasite lines: the first was WT-ANKA (a wild type parasite that does not express GFP), and the other was WT-GFP (WTGFPcon 507

cl1), a line in which green fluorescent protein is expressed constitutively (Janse et al, 2006).

The kinesin8B-GFP C-terminal tagged transgenic line was generated by modification of the endogenous gene in the WT-ANKA parasite. This transgenic line develops normally at various stages of development, as is now mentioned on page 14, line 326 and in the new Supplementary Table 2.

The kinesin-8B gene knockout line was generated from the WT-GFP line and it is to be noted that kinesin-8B is not GFP-tagged in this line. This point has now been clarified in the manuscript on page 22, line 510.

We apologise for this confusion due to lack of clarity.

Reviewer #2 (Comments to the Authors (Required)):

The authors describe the functional involvement of kinesin-8B in formation of axonemes in Plasmodium by gene deletion and gene tagging.

As such this is a straightforward and well written and nicely illustrated paper that could in principle be published as is. However, some small additions would really improve the paper. I would have indicated some minor spellings as well if there were line numbers.

We thank the reviewer for these positive comments. We have incorporated the additions suggested by the reviewer, as described below and in the manuscript on pages 10, 11, 14, 17, 18, 22, 23 and 42. The spelling has been checked and errors corrected. We have also included number line to the manuscript.

Is the kinesin-GFP only visible in sexual stages or also in other stages along the life cycle? If yes, please show, if just in supplement.

We confirm that there is no detectable expression of kinesin-8B at any other stage of life cycle. This is mentioned in the revised manuscript on page 14, line 322, and we have added comparative images in a new Supplementary Figure 3 (Fig S3).

If the authors have electron microscopy data from later time points (e.g. 30 minutes) on the mutants I would encourage them to show this. Maybe flagella do assemble to some degree and the RNAseq analysis was performed at this time.

We wish to clarify that electron microscopy data were obtained at both 15 and 30 min time points. The appearance of the mutant was similar at both time points; there was no evidence of axoneme formation in the mutant at either 15 or 30 min. We have modified the text to clarify that the observations are representative of both 15 and 30 min time points for the mutant and no differences were noted.

Can the authors quantify the amount of microtubules in the KO vs the WT? It appears that the microtubules are just as long in the KO as in the WT and that there are just as many microtubules present, is that impression correct?

We can't say anything about total microtubule length, as complete microtubules are not included in a single section. However, even though the total length cannot be measured, both WT and mutant parasites appear to show similar microtubule elongation. We have included quantitative data on the incidence of normal and abnormal axonemes in the WT and the absence of axoneme formation in the mutant. In addition we measured the relative number of basal bodies and compared their relationship to the nuclear poles in both WT and mutant parasites. The electron microscopy results section has been rewritten to include this information.

What is the localization of Ndc80 in other parasites stages (this might be part of a separate paper and could be stated as such, alternatively a figure with different life cycle stages would be informative).

We appreciate this comment by the reviewer about the location of Ndc80 in other parasite stages. Ndc80 is expressed at every proliferative stage of the parasite. A manuscript describing a parasite line that expresses a fluorescent tagged Ndc80 is nearing completion and will be uploaded onto BioRxiv shortly.

It should be clearly stated in the figure legend if these parasites were not clonal lines.

We have addressed this comment on page 39.

Some references are missing page or e or doi numbers

We have checked and updated the bibliography, adding the two new references suggested by the reviewer.

The discussion is not very eloquent and could be improved by subheadings and including a small discussion on two different aspects:

1. Motility of the gametes: Plasmodium gametes move in a very distinct way from other flagella and only one paper has really examined this in detail (Wilson et al., PNAS 2010). This would deserve mention and some discussion as I think no paper on gametes can be complete without this landmark study (could also be mentioned in the introduction)

We thank the reviewer for this suggestion. We have tried to improve the eloquence of the discussion but found that the text flowed better in this section without subheadings. We have now included information on the motility of the gametes in the discussion section, and have added the relevant citation of Wilson et al, PNAS 2013 on page 17.

2. Dissociation of nuclear division with gamete formation is reminiscent of a similar phenotype observed after microtubule depletion in oocysts, where nuclear division still occurs (albeit in the absence of microtubules) but no sporozoites are formed (Spreng et al., EMBO J 2019). This, together with the author's observation on kinesin 8B and the two drugs suggests a distinct molecular makeup and functionality of the different microtubule arrays in the parasite.

We agree with reviewer. The microtubules (MTs) have distinct location and function in different stages of the parasite life cycle. In gametocytes, it is very clear that axoneme assembly happens in the cytoplasm and the MTs that are recruited here are very distinct from nuclear spindles. This has now been included in the discussion on page 18.

August 2, 2019

RE: Life Science Alliance Manuscript #LSA-2019-00488-TR

Dr. Rita Tewari
University of Nottingham
School of Life Sciences
Queens Medical Centre
Nottingham NG7 2UH
United Kingdom

Dear Dr. Tewari,

Thank you for submitting your revised manuscript entitled "Kinesin-8B controls basal body function and flagellum formation and is key to malaria transmission". I have now assessed the revision performed and appreciate the introduced changes. I would thus be happy to publish your paper in Life Science Alliance pending final minor revisions:

- Reviewer #1 was also concerned by the effects the GFP tag could have on Kinesin-8B localization, please add a short discussion on this point
- please note that you currently use Table S1 and video SV1 twice, please fix
- fig S3 was flagged up in our routine image check, please provide source data for this figure

A. FINAL FILES:

-- Summary blurb (enter in submission system): A short text summarizing in a single sentence the study (max. 200 characters including spaces). This text is used in conjunction with the titles of

papers, hence should be informative and complementary to the title. It should describe the context and significance of the findings for a general readership; it should be written in the present tense and refer to the work in the third person. Author names should not be mentioned.

B. MANUSCRIPT ORGANIZATION AND FORMATTING:

Sincerely,

We are very pleased that the Editor and reviewers are happy to publish our paper in Life Science Alliance pending final minor revisions. We have addressed remaining comments in this revised manuscript. We hope that our revised manuscript is now acceptable for publication. A point-by-point response to the comments from this second revision is provided below and the modifications are highlighted in red in the revised manuscript.

- Reviewer #1 was also concerned by the effects the GFP tag could have on Kinesin-8B localization, please add a short discussion on this point.

We have included the statement describing the effects caused by GFP tagging of kinesin-8B to parasite growth on page 12 and providing data in a supplementary table (Table S2).

- please note that you currently use Table S1 and video SV1 twice, please fix

We apologise for the mistakes. These are now corrected.

- fig S3 was flagged up in our routine image check, please provide source data for this figure

We have followed all the guidelines to generate the figure and also providing the source data for this figure both in power point and PDF format. Hope this is fine now.

August 6, 2019

RE: Life Science Alliance Manuscript #LSA-2019-00488-TRR

Prof. Rita Tewari
University of Nottingham
School of Life Sciences
Queens Medical Centre
Nottingham NG7 2UH
United Kingdom

Dear Dr. Tewari,

Thank you for submitting your Research Article entitled "Kinesin-8B controls basal body function and flagellum formation and is key to malaria transmission". It is a pleasure to let you know that your manuscript is now accepted for publication in Life Science Alliance. Congratulations on this interesting work.

DISTRIBUTION OF MATERIALS:

Again, congratulations on a very nice paper. I hope you found the review process to be constructive and are pleased with how the manuscript was handled editorially. We look forward to future exciting submissions from your lab.

Sincerely,

Andrea Leibfried, PhD
Executive Editor
Life Science Alliance
Meyerohofstr. 1
69117 Heidelberg, Germany
t +49 6221 8891 502
e a.leibfried@life-science-alliance.org
www.life-science-alliance.org